# One dimensional wormhole corrosion in metals

Yang Yang [1,2,10] ✉, Weiyue Zhou [3,10], Sheng Yin [4], Sarah Y. Wang [5], Qin Yu [4], Matthew J. Olszta [6], Ya-Qian Zhang [5], Steven E. Zeltmann [5], Mingda Li [3], Miaomiao Jin [7], Daniel K. Schreiber [6], Jim Ciston [1], M. C. Scott [1,5], John R. Scully [8], Robert O. Ritchie [4,5], Mark Asta [4,5], Ju Li [3,9], Michael P. Short [3] ✉ & Andrew M. Minor [1,4,5] ✉

Corrosion is a ubiquitous failure mode of materials. Often, the progression of localized corrosion is accompanied by the evolution of porosity in materials previously reported to be either three-dimensional or two-dimensional. However, using new tools and analysis techniques, we have realized that a more localized form of corrosion, which we call 1D wormhole corrosion, has previously been miscategorized in some situations. Using electron tomography, we show multiple examples of this 1D and percolating morphology. To understand the origin of this mechanism in a Ni-Cr alloy corroded by molten salt, we combined energy-filtered four-dimensional scanning transmission electron microscopy and ab initio density functional theory calculations to develop a vacancy mapping method with nanometer-resolution, identifying a remarkably high vacancy concentration in the diffusion-induced grain boundary migration zone, up to 100 times the equilibrium value at the melting point. Deciphering the origins of 1D corrosion is an important step towards designing structural materials with enhanced corrosion resistance.

Corrosion[1] is a notorious problem leading to early failure and increased cost to mitigate for engineering systems in aircraft, bridges, and nuclear reactors[2–5] as well as functional devices such as batteries, sensors, and biomedical implants[6–8]. While some classical theories[9,10] predict a uniform corrosion process, corrosion is often accelerated at specific sites due to various types of material defects and distinct local environments[11–13]. Accelerated localized corrosion can proceed insidiously for years as it is intrinsically more difficult to detect. Thus, localized corrosion presents a greater threat for component failure than uniform corrosion, especially when coupled with stress leading to

processes such as stress-corrosion and sulfidation-induced cracking[5,13,14]. However, the detection, prediction, and study of localized corrosion are extremely challenging, as the length-scale of incubation sites is very small, typically at the nanometer scale[11,15] and below, beyond limits of non-destructive detection. Traditional high-resolution imaging techniques struggle to efficiently capture the phenomenon in its early stages due to multiple issues such as a small, non-representative field of view, the limits of imaging across time-scales, the inability to recover three-dimensional (3D) information from 2D images, and the undesired modification of source materials

[1]National Center for Electron Microscopy, Molecular Foundry, Lawrence Berkeley National Laboratory, Berkeley, CA, USA. [2]Department of Engineering Science and Mechanics and Materials Research Institute, The Pennsylvania State University, University Park, PA, USA. [3]Department of Nuclear Science and Engineering, Massachusetts Institute of Technology, Cambridge, MA, USA. [4]Materials Sciences Division, Lawrence Berkeley National Laboratory, Berkeley, CA, USA. [5]Department of Materials Science and Engineering, University of California, Berkeley, CA, USA. [6]Energy and Environment Directorate, Pacific Northwest National Laboratory, Richland, WA, USA. [7]Department of Nuclear Engineering, The Pennsylvania State University, University Park, PA, USA. [8]Department of Materials Science and Engineering, University of Virginia, Charlottesville, VA, USA. [9]Department of Materials Science and Engineering, Massachusetts Institute of Technology, Cambridge, MA, USA. [10]These authors contributed equally: Yang Yang, Weiyue Zhou. ✉e-mail: yang@psu.edu; hereiam@mit.edu; aminor@berkeley.edu

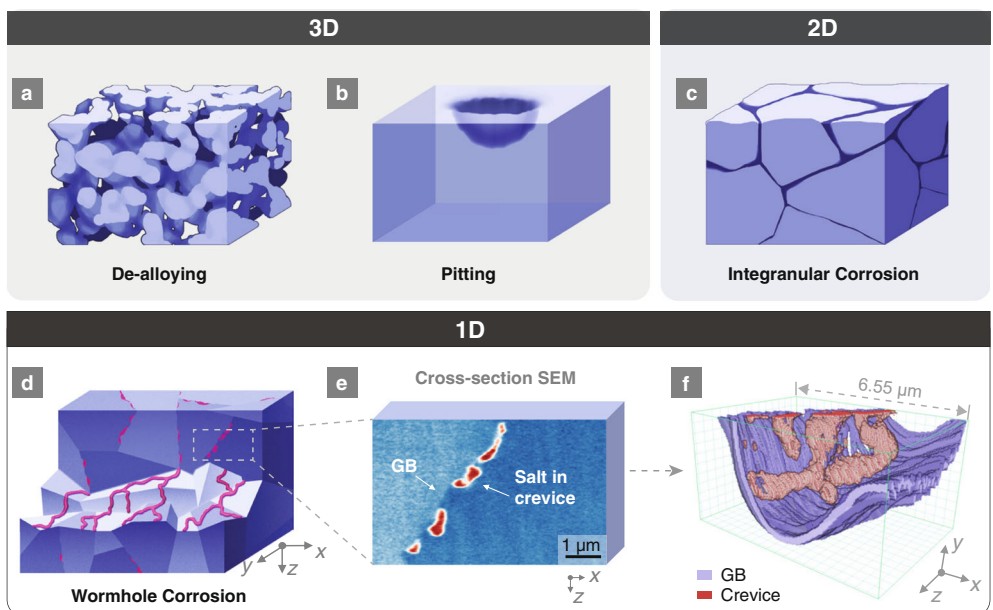

**Fig. 1 | Differences between 3D, 2D, and 1D corrosion.** All corrosion occurs in a top-down manner, i.e., the penetration direction through thickness is from −z to +z. **a** Schematic drawing of a typical bulk bicontinuous dealloying corrosion morphology. **b** Schematic drawing of a typical pitting corrosion morphology with the volume set as transparent. The grain boundaries (GBs) were not displayed because both intergranular and intragranular pitting can occur. **c** Schematic drawing of a typical intergranular corrosion morphology with the volume set as opaque. The sides of the cube display the cross-sectional views. **d** Schematic drawing of 1D wormhole corrosion in a polycrystalline material. The upper half is a cross-sectional view showing discontinuous dots (i.e., voids filled by molten salt) along the GBs. The bottom half is a volumetric cutaway along the GBs where the 1D percolating network of tunnels on the grain surfaces is clearly shown in red. The dark blue color in **a** to **c** indicates free space such as cracks, voids, crevices etc., while the red color in **d** indicates wormholes filled with molten salt. **e** A representative false-colored SEM image showing the cross-section of Ni-20Cr after corrosion. **f** FIB-SEM 3D reconstruction of the volume shown in (**e**). The dimension of the box is 6.6 μm (x) × 3.5 μm (y) × 4.1 μm (z).

from sample preparation. The lack of understanding of localized corrosion has further added to the uncertainty in engineering systems, especially in newer, less-studied corrosive environments. A prominent example of such an environment is molten salt, which has become increasingly important as a reaction medium for materials synthesis[16,17], a solvent for materials recycling[18,19], and a coolant/fuel for next-generation nuclear reactors[20,21] and concentrated solar power (CSP) plants[22,23].

Molten salt systems typically operate at high temperatures (350–900 °C) and require structural materials that can withstand these extreme conditions for years to decades. Ni or Fe-based alloys are considered strong candidates for high-temperature applications and molten salt next-generation nuclear reactors. In particular, these alloys are being tested as vessel and primary circuit structural materials for both nuclear and CSP plants, critical for containing the hot molten salt, and, in the case of some reactor designs, the molten nuclear fuel. Previously, it has been shown that the corrosion of these alloys in molten fluoride/chloride salt proceeds via the preferential leaching of the most electrochemically susceptible element (usually Cr) into the salt, which appears to create voids in the metal. Early studies proposed that these voids were discrete Kirkendall voids[24–26], or remnants of precipitates that were preferentially attacked during corrosion[27]. However, more recent research has shown strikingly that salt exists in some of these voids[23,28,29], and can even lead to through-material penetration[30], posing a critical question on how these voids form and mediate salt infiltration into the metal. The answer to this question is the key to the fundamental understanding of localized corrosion in molten salt environments, yet it still remains unclear.

Here we experimentally show and mechanistically explain the origin of a form of highly localized penetrating corrosion in a Ni-20Cr alloy attacked by molten fluoride salt, using a combination of correlative electron microscopy and atomistic simulation. We term this phenomenon *1D wormhole corrosion*, named not only for its wormhole-like 1D morphology but also because it can function as a mass-flow pathway (in contrast to diffusional pathways), which we find is responsible for the extremely rapid infiltration of molten salt. This mechanism creates an extremely localized corrosion morphology (Figs. 1 and 2) that greatly increases the depth of penetration per volume of corroded metal when compared to other corrosion morphologies.

## Results

Figure 1 schematically shows this unique 1D void morphology, alongside comparisons to previously known localized corrosion processes such as pitting[15,31] and intergranular corrosion[32,33]. As the directions for void growth in bicontinuous dealloying[34] (Fig. 1a) and pitting (Fig. 1b) incur no constraints, they can be described as 3D corrosion morphologies. Another well-known mechanism of void penetration is intergranular corrosion (Fig. 1c), which is confined to two-dimensional (2D) grain boundaries (GBs). In contrast, the concept of a 1D and percolating penetration through thickness/depth, such as the wormhole corrosion depicted in Fig. 1d-f, has not been documented in any open literature. In 1D wormhole corrosion, void tunnels with a high aspect ratio are constrained to grow along selected routes on GBs without fully covering the GB plane, establishing a "capillary-like" percolating system along GBs. We describe it as "1D" because GBs are regarded as a 2D percolating transport network, while the 1D wormhole corrosion builds a 1D percolating network constrained by the 2D GB network. The wormholes observed here reveal the routes that facilitate the ingress of the corrosive fluid to enable much more rapid and sustained corrosion.

When a cross-section of a GB with 1D corrosion is made, the voids along the GB manifest themselves as discontinuous "dots" (as shown in the boxed region in Fig. 1d, e), which may easily be mistaken as disconnected voids. By comparison, a cross-section of a GB with 2D corrosion would result in a continuous line of voids (Fig. 1c). As such, a

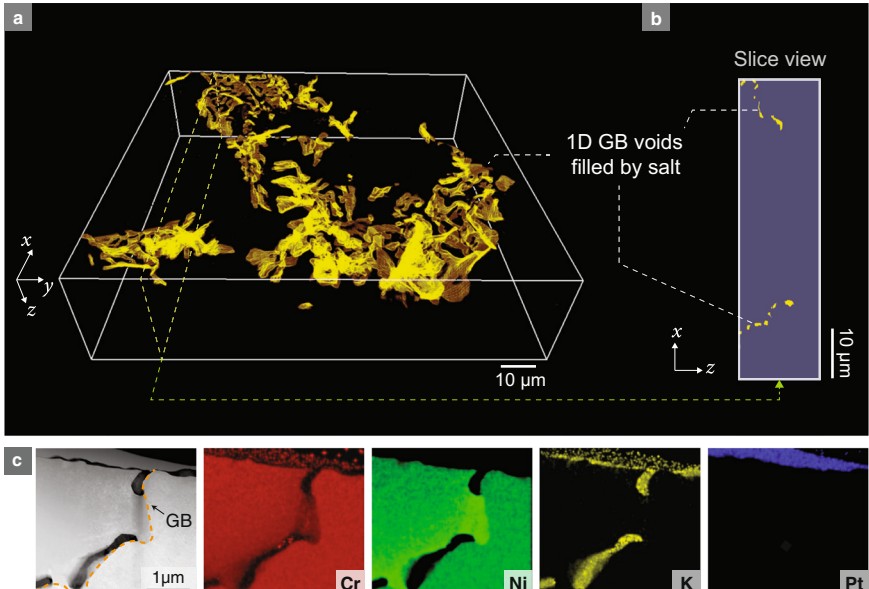

**Fig. 2 | Evidence of 1D wormhole corrosion and preferential etching/corrosion along one side of the grain boundary (GB).** The primary salt penetration/corrosion direction is along the *z* axis. **a** FIB-SEM 3D reconstruction of a much larger volume in a Ni-20Cr sample after molten salt corrosion. **b** A single-slice view showing discontinuous voids along GBs. **c** STEM HAADF image and EDX chemical mapping depicting elemental distributions from a thin (<100 nm) slice of the GB corrosion microstructure on a representative Ni-20Cr sample.

convenient way to differentiate 1D, 2D, and 3D corrosion is to produce a cross-section, observe the void morphology, and determine whether the externally corroding fluid exists in the voids. Figure 1e shows an example of this cross-sectional evidence from 1D corrosion in a Ni-20Cr alloy foil after exposing one side to fluoride molten salt at 650 °C. The experimental setup used is schematically illustrated in Supplementary Fig. 1. We found that while the samples appeared to be intact after 4 h of corrosion, the molten salt had already fully penetrated the 30-μm-thick Ni-20Cr foil via 1D wormhole corrosion, as evident in the scanning electron microscopy (SEM) image showing molten salt at the opposite side of the sample (Fig. 3d). To further confirm this 1D infiltration, we used argon ion-milling to prepare a cross-section (see Supplementary Fig. 1), and we performed correlative electron microscopy characterization on the surface of the cross-section. Figure 1e shows a typical SEM image (false-colored) of the cross-sectioned surface depicting several discrete voids. While the voids appear discontinuous along the GB, they are expected to be connected via linked voids on different planes based on our 1D model and the presence of salt within. To verify this, the corresponding region in Fig. 1e is serially sectioned with a focused ion beam (FIB) and imaged with the SEM to create a 3D reconstruction. The resulting reconstruction, shown in Fig. 1f and Supplementary Movie 1, proves the connectivity of the voids along the GB, forming a network of 1D tunnels that facilitate molten salt infiltration. Another 3D reconstruction of a much larger volume is presented in Fig. 2a, b and Supplementary Movie 2, showing the percolating network of 1D wormholes in another representative sample and the reproducibility of this experiment. A FIB sample lifted out near the center of the cross-section surface (see Supplementary Figure 1e) along the GB is characterized by scanning transmission electron microscopy (STEM) and energy-dispersive X-ray spectroscopy (EDX). The presence of potassium in the voids demonstrates the existence of salt in the crevices along the GBs, further supporting our observation that 1D percolating "wormholes" facilitate rapid salt penetration. The salt is likely wicked by capillary forces into the open voids, and must be present at the head in electrochemical systems. It should be noted that the morphology of discontinuous voids along GBs can also be found in the cross-sections of other systems with intergranular voids due to radiation damage[35], Kirkendall effects[36], or etched precipitates[37].

However, 1D corrosion is differentiated by the fact that the voids are interlinked and form a percolating network. Filiform corrosion[38,39] can also produce similar 1D and percolated channels, but these surface effects are confined to specific metal/organic-film interfaces, and therefore coupled with a thin film delamination process. The 1D-like channel formation underneath the organic coatings during filiform corrosion is not considered a bulk corrosion (*i.e.*, depth penetration) mechanism like 1D wormhole corrosion. Also, the 1D wormhole corrosion is significantly different from other "1D" corrosion morphologies reported in previous literatures (see more discussions in the Supplementary Discussion).

Pore structure is a distinguishing feature of localized versus uniform corrosion, as it dictates enhanced, local depth-penetration in materials. The more confined the void structure, the higher the rate of penetration per unit volume corroded, and the more detrimental its effects. While "finger-like" intergranular oxide protrusions[40,41] have been observed before, they are only found localized at the corrosion front (i.e., deepest part of the oxidized region) serving as the terminus of 2D corrosion instead of as a global feature. Here, the wormhole corrosion that we discovered is dominated by 1D-features, suggesting that the morphology is stable despite being extremely localized. The penetration efficiency of 1D corrosion is further amplified by the lack of passivation mechanisms, as the corrosion products, once formed, can directly dissolve into the molten salt present in the 1D wormholes. By contrast, the advancement of a corrosion front along the depth direction in aqueous environments is more sluggish because the passivation layer needs to form and break repeatedly; this a typical characteristic of stress-corrosion cracking (SCC)[32]. As such, if we define the penetration efficacy as the maximum depth of corrosion divided by the total mass-loss of metal, then 1D wormhole corrosion manifests a remarkable penetration efficacy even without an externally applied stress. Such depth-focusing penetration morphology implies the need for prohibitively thick structural components to fully encapsulate the radioactive fuel during the long-term reactor operation if the 1D form of corrosion occurs and persists. Meanwhile, these percolating voids, even before complete penetration, can readily impact the mechanical performance of structural components via

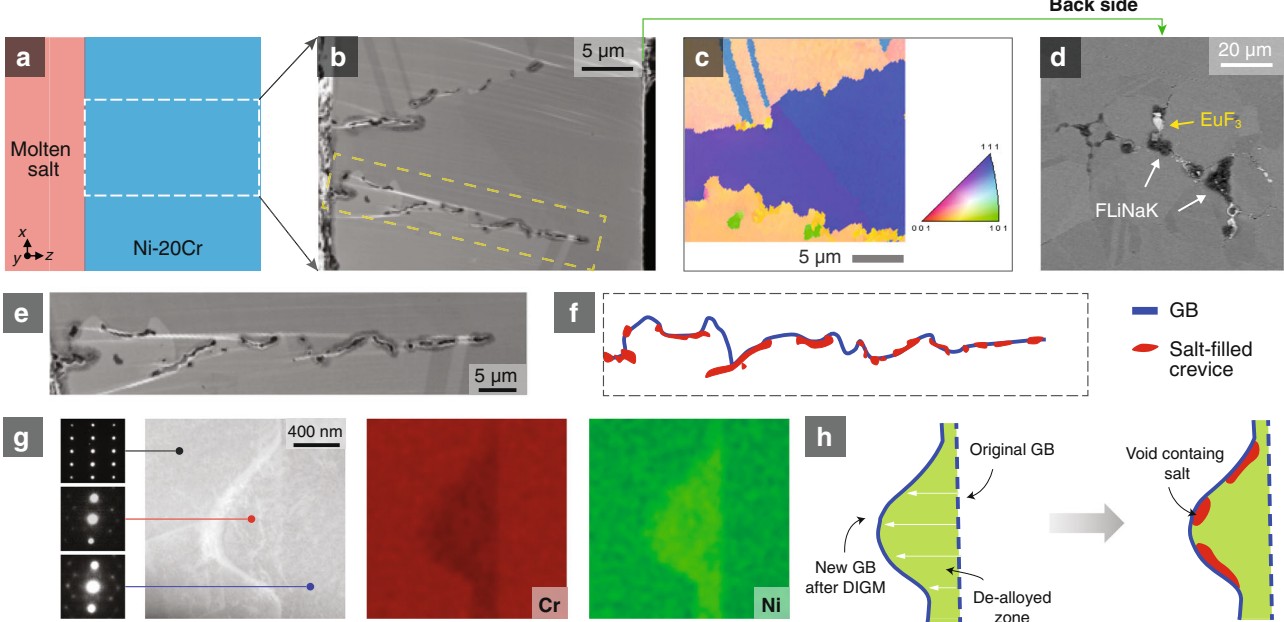

**Fig. 3 | Preferential growth of wormholes in the diffusion-induced grain boundary migration (DIGM) zone. a–c** EBSD evidence showing rugged grain boundaries (GBs) after DIGM. **d** Evidence of salt penetration: SEM imaging shows that salt exists on the backside of the 30-μm-thick foil sample. **e** Representative SEM image corresponding to the boxed area in (**b**), which shows a rugged GB after molten salt corrosion. **f** Schematic drawing of the key features in (**e**). **g** STEM-HAADF image, STEM-EDX mapping, and SAED evidence for DIGM. **h** Schematic illustration of the DIGM process and the preferential growth of wormholes in the DIGM zone.

GB embrittlement or strain localization near voids[13]. The implication of this mechanism calls for research and development of alloy design strategies to mitigate 1D wormhole corrosion in molten salt environments.

In addition to the 1D nature of the molten salt infiltration, we also observed several other intriguing morphological phenomena: specifically, the asymmetric growth of voids and a Cr-depleted zone along GBs. During molten salt corrosion, Cr is preferentially attacked due to the lower free energy of formation of Cr fluorides compared to those of Ni and Fe. As such, Cr leaches out from the bulk, diffuses to the GB, and rapidly transports to the metal-salt interface to react with the salt[30]. The selective dissolution of Cr results in the dealloying of the underlying metal. Previously, it has been shown by SEM-EDX that Ni is enriched while Cr is depleted along GBs after molten salt corrosion[30]. Here, we use high-resolution STEM-EDX mapping to show that this Cr depletion is also asymmetric with respect to the GB. As shown in Fig. 2c and Fig. 3g, the Ni-enriched, Cr-depleted zone prefers to grow on one side of the GB such that the GB is tangent to this dealloyed zone. As diffusivity is relatively isotropic in face-centered cubic (FCC) Ni-20Cr, the asymmetric dealloying with respect to the GB is unlikely to result directly from different grain orientations on either side of the GB. Instead, further characterization of the cross-section suggests that the GB has migrated during corrosion. Figures 3a–c and 3e, f show a representative GB after corrosion in molten salt that is highly curved, verified by electron backscattered diffraction (EBSD) orientation mapping (Fig. 3a–c). This is unexpected, as prior to corrosion GBs tend to be much smoother in order to minimize their surface energy, particularly in large grained materials such as the one used in this study. The rugged structure of the GB is caused by the heterogeneous nature of GB migration during corrosion. This observation agrees with the phenomenon of diffusion-induced grain boundary migration (DIGM) previously reported in alloying/dealloying diffusional systems[42] and SCC of steels in water[43–47].

Figure 3h and Supplementary Movie 3 schematically illustrate the detailed process of DIGM. The rapid diffusion of Cr along GBs to the metal-salt interface induces the migration of the GB from the right to

the left, and the area that the GB sweeps through becomes a locally dealloyed zone. This dealloyed zone shares the same crystal orientation as the grain on the right, yet it displays a very different composition. This DIGM process is confirmed by our STEM-EDX and TEM selected area electron diffraction (SAED) characterizations shown in Fig. 3g. The effects of GB migration on the speed of corrosion are considered to be two-fold. First, the curved GBs increase the total length of salt infiltration pathways, slowing down the penetration. On the other hand, the DIGM process will modify the local GB inclination gradually. If there is a preferential inclination for fast penetration, such a process will steer the salt front to this fast etching route along GBs, facilitating faster penetration. Since we observe such fast penetration in our experiments, the latter (i.e., preferential GB inclinations for fast penetration) appears to be the dominant process. Further discussion is presented in the supplementary materials with a Monte Carlo simulation (Supplementary Note 1 and Supplementary Movie 4). It is hypothesized that DIGM is not necessary for the formation of wormholes, yet it can impact the kinetics of wormhole formation.

Similar to the dealloyed zone, the 1D crevice shows asymmetry along GBs, as is evident in Figs. 2c and 3e, f. A careful inspection of TEM samples extracted from different locations reproducibly shows that these crevices always reside on the Cr-depleted side of new GB positions through the DIGM zone (see schematic in Fig. 3h), suggesting an intrinsic correlation. Since there is a reduction of more than 5 at% Cr in the dealloyed zone, we hypothesize that these DIGM zones should be rich in vacancies and vacancy clusters, and thus can act as precursors for void formation. However, the vacancies are too small to be captured by traditional high-resolution electron microscopy techniques. To measure the vacancy supersaturation in the DIGM zone, we developed a multimodal method combining energy-filtered four-dimensional scanning transmission electron microscopy (4D-STEM) lattice parameter mapping[48], EDX elemental mapping, and density functional theory (DFT) modeling. By combining these techniques, it is possible to analyze the local vacancy concentration qualitatively with nanometer resolution, achieving a spatial resolution at least $10^5$ times higher than that of conventional approaches such as positron annihilation

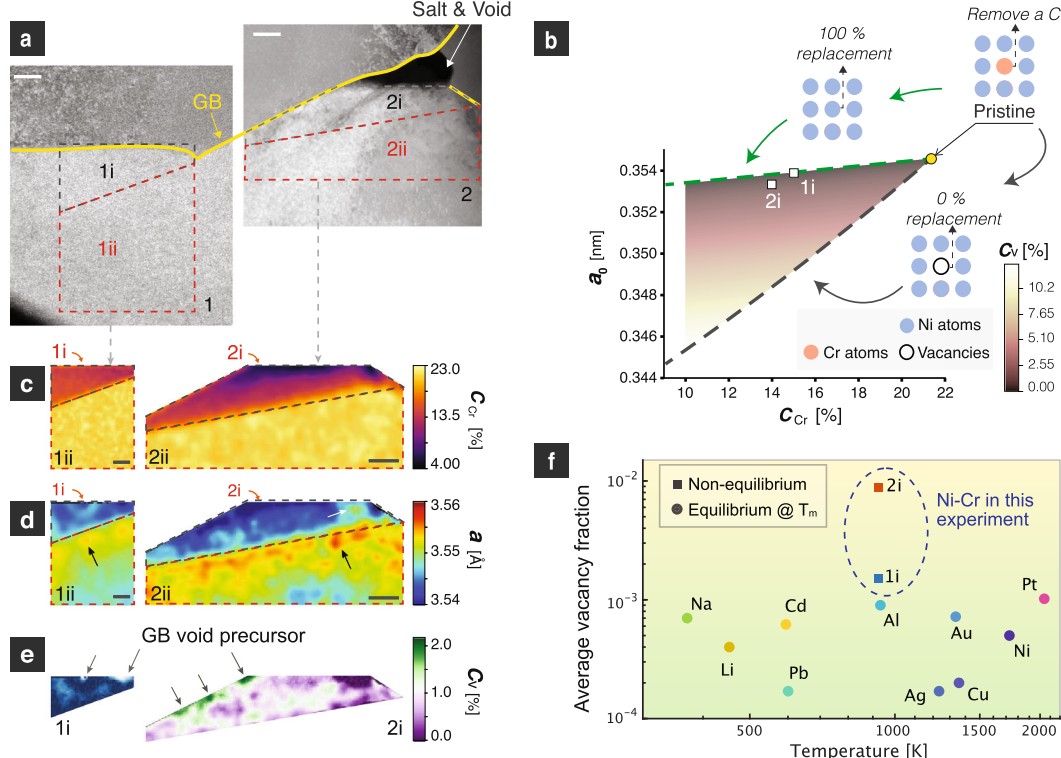

**Fig. 4 | Evidence of wormhole precursors in the diffusion-induced grain boundary migration (DIGM) zone indicated by nanometer-resolution vacancy mapping. a** STEM-HAADF image showing two selected zones with zone 2 being closer to the salt. **b** DFT simulations of the phase map of the dealloying process, indicating the correlation between the relaxed lattice constant ($a_0$), vacancy fraction ($C_v$), and Cr fraction ($C_{Cr}$). **c** Mapping of Cr fraction by STEM-EDX. **d** Mapping of lattice parameters by 4D-STEM. **e** Mapping of vacancy fraction in the DIGM zones. **f** Comparison of the vacancy fraction in different systems. The measured vacancy fractions in the DIGM zones (1i and 2i) and in pure metals at equilibrium close to the melting temperature $T_m$ are plotted and compared. The data for the pure metals are from refs. [74–77]. All scale bars are 200 nm.

spectroscopy (PAS). With this technique, we show that the DIGM zones possess a massive vacancy supersaturation and that the DIGM zones closer to the salt-filled voids tend to have a higher vacancy concentration (Fig. 4), in agreement with our hypothesis. We also found discrete vacancy "hot spots," hypothesized to be the precursors of wormholes, localized near a GB. The vacancy mapping analysis is detailed below.

Figure 4a shows a representative STEM high-angle annular dark-field (HAADF) image of another TEM sample lifted out from the cross-section of the Ni-20Cr foil. Regions 1 and 2 refer to two different regions below a GB, where region 2 is closer to the salt in the crevice. Figure 4c shows the elemental mapping of the Cr fraction in regions 1 and 2, where the DIGM zones (regions 1i and 2i) are distinguished from the non-DIGM zones (regions 1ii and 2ii) by local Cr fractions. Figure 4d presents the 4D-STEM lattice spacing maps in these two regions, where a significant reduction in lattice spacing is shown in the DIGM zones (regions 1i and 2i). This decrease in lattice spacing can be interpreted as a type of phase transformation induced by the changes in Cr and vacancy fractions while the crystallographic symmetry is maintained. In this vein, the dealloying process can be depicted in a phase map that can be calculated by DFT simulations. This phase map, as shown in Fig. 4b, describes the intrinsic relationship between the relaxed lattice constant ($a_0$), the Cr fraction ($C_{Cr}$), and the vacancy fraction ($C_v$). Starting from pristine Ni-20Cr (upper right corner of the triangle in Fig. 4b), the dealloying process shifts the DIGM zone's position within the triangular phase map by altering $C_{Cr}$ and $C_v$. The upper line of this triangle represents "100% replacement," meaning that each Cr atom removed is substituted with a Ni atom. Similarly, the bottom boundary of the triangle represents 0% replacement, indicating that no atom will refill the lattice site when a vacancy is created. One should also note that the relaxed lattice constant $a_0$ can be related to the actual lattice

spacing $a$ and the elastic strain $\varepsilon^e$ by

$$\varepsilon^e = (a - a_0)/a_0 \qquad (1)$$

where $a$ and $C_{Cr}$ can be measured by 4D-STEM and EDX, respectively. If either $\varepsilon^e$ or $C_v$ can be measured, then the other can be derived. We show that by measuring the strain in the non-DIGM zone, we can estimate the average $\varepsilon^e$ within the DIGM zone using the theory of Eshelby's inclusion (Supplementary Note 2 and Supplementary Figs. 3–4). This enables the back-calculation of $C_v$ in the DIGM zones. The vacancy mapping results corresponding to regions 1i and 2i are shown in Fig. 4e, and the region-averaged vacancy fractions are plotted in Figs. 4b, 4f and Supplementary Fig. 5, indicating a higher vacancy fraction in 2i than that in 1i, which is reasonable as 2i is much closer to the salt thus the driving force for the outward diffusion of Cr is more significant. The vacancy "hot spots" (indicated by arrows in Fig. 4e) are shown to be localized near the GB. The discrete nature of these "hot spots" agrees with our observation that the wormholes appeared discontinuous along a GB in a 2D image, suggesting that these "hot spots" are precursors of wormholes. The averaged vacancy fraction is also estimated to be ~9 × 10⁻³ in region 2i, which is about 10 to 100 times greater than the equilibrium values for metals close to their melting points, and about 1000 to 10,000 times that found in their equilibrium counterparts at 650 °C (Fig. 4f). The supersaturation of vacancies found in these DIGM zones is a result of the rapid leaching of Cr, which drives the system out of equilibrium. The excess vacancy concentration is consistent with our observation that the voids and DIGM zone are found on the same side of the GB, strengthening our hypothesis that the dealloyed zone is a precursor to void formation. Accordingly, the previous DIGM theory by Broeder[49] is revised and

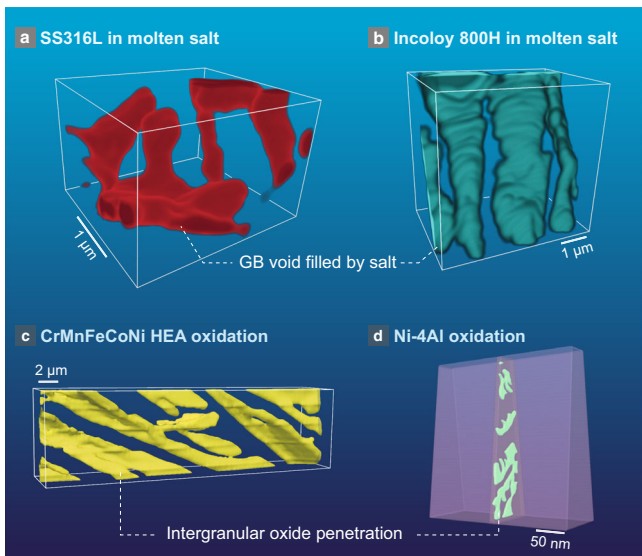

**Fig. 5 | Electron microscopy 3D reconstruction showing 1D penetrating corrosion in other systems. a** FIB-SEM reconstruction of a 316 L stainless steel (SS) sample after corrosion in molten salt. **b** FIB-SEM reconstruction of an Incoloy 800H sample after corrosion in molten salt. **c** FIB-SEM reconstruction of a CrMnFeCoNi high-entropy alloy (HEA) sample after oxidation at 1000 °C. **d** TEM tomography reconstruction of a Ni-4Al sample after oxidation in water at 330 °C.

presented in Supplementary Note 3 and Supplementary Fig. 6 to account for the excess free volume in the DIGM zone.

While the vacancy supersaturation in the DIGM zone provides precursors for void nucleation, the growth of void in the lateral direction (perpendicular to the GB) is a self-limiting process. Re-passivation by self-healing of the oxide layer cannot be a mechanism in these wormholes, although it is in pits and cracks in aqueous environments. Since the Cr concentration near the sidewall is very low (<5%) and the Ni concentration is very high, this Ni enriched-layer servers as a barrier that slows down the lateral growth of each void. Considering that the wormholes prefer to focus on depth penetration instead of growing laterally, we hypothesize that most of the anodic reactions occur at the heads of wormholes, while the internal surfaces of the tunnels may provide the area for cathodic reactions.

Strain localization at interfaces due to corrosion is often detrimental and can catalyze mechanical instability and failure. We also present here strain mapping in and outside the DIGM zones. From Fig. 4d and Supplementary Fig. 3e, f, we find that the strain in these DIGM regions is tensile, and the local strain maximum ranges from 0.3 to 3%. Meanwhile, there is an appreciable amount of strain localized near the void (indicated by the white arrow in Fig. 4d), suggesting a pathway vulnerable to crack propagation. We also noticed that there is a tensile strain (indicated by the black arrows in Fig. 4d and Supplementary Fig. 3d) surrounding the DIGM zones inside the matrix, agreeing with our Eshelby's inclusion analysis (Supplementary Note 2) that the DIGM zone tends to shrink due to vacancy enrichment and Cr depletion.

In this study, a series of advanced electron microscopy techniques including FIB, SEM, 3D tomography, SAED, and HAADF-STEM-EDX characterization are combined to discover and explain the mechanism of 1D wormhole corrosion in Ni-20Cr during exposure to high-temperature fluoride molten salt. We also developed an approach to perform nanometer-resolution vacancy mapping by combining energy-filtered 4D-STEM lattice parameter mapping and EDX with DFT simulations, revealing a remarkably high vacancy concentration to be responsible for void nucleation and 1D growth. This, in turn, allowed us to relate the asymmetric void formation to the excess vacancies in the DIGM zone. This extreme form of localized wormhole corrosion has a

remarkably high mass-specific penetration capability among different corrosion mechanisms, yet it is far too localized to easily detect, making it a critical potential threat to structural materials which must be mitigated. We have conducted tomographic imaging experiments on several other corrosion systems and found similar 1D penetrating corrosion morphologies (Fig. 5 and Supplementary Movies 5–8), suggesting that 1D-type corrosion may not just be an anomaly of the molten salt environment. It is intriguing that intergranular oxide can also form a percolated network similar to the 1D wormhole corrosion (Fig. 5c, d). However, it still requires further in situ 3D observation to validate whether these oxide networks share the same mechanism as 1D wormhole corrosion, as there exist two possibilities: (1) The oxide grows in a similar way to wormhole corrosion; (2) The oxides nucleate first and then link together (like the linkage of intergranular carbides due to sensitization). The former would be true if the principal pathway of oxygen transport were to be the metal-oxide interface or the oxide itself, while the latter would be true if the metal-metal interface were to be the dominant oxygen transport pathway. The reality could even be a mixture of both cases. In fact, we believe it is probable that 1D wormhole corrosion is a common, yet hitherto largely unrecognized, type of corrosion mechanism in both molten salt environments and high-temperature corrosion reactions such as sulfidation-oxidation or flux that involves mass transport through liquids[14]. However, because of the lack of dry-polishing methods such as Ar+ milling that preserve the salt-filled holes, the absence of 3D FIB-SEM tomography to reveal the internal percolation, and the complexity arising from intergranular precipitates in commercial alloys[26,50–54], this phenomenon was not realized until this current study.

As a consequence of 1D corrosion, strategies must now be developed to slow, prevent, or stop wormhole-type corrosion in order to increase the longevity and safety of next-generation nuclear reactors and concentrated solar power plants. The deep infiltration of salt into metal also indicates that the corrosion front is hidden deep inside the bulk, where the local chemistry can be significantly different from that near the bulk surface. As such, modeling and experimental efforts should account for the unique local chemistry inside the wormholes in order to better understand their corrosion behavior and protect against it. Also, temperature has recently been found to play a critical role on the corrosion morphologies in molten salt[55]. Understanding how temperature interacts with 1D wormhole corrosion, especially the temperature range where 1D wormhole dominates, would offer valuable insights into mitigating it in structural materials. Last but not least, corrosion has been recently utilized as a method to produce unique void structures, such as 1D nanotubes[56] and 3D bi-continuous structures[34,57], for functional applications such as catalysis or sensors. While we have identified this mechanism as occurring in a molten salt environment, one might envision similar scenarios in other dissolution-driven systems with the right balance of dissolution rates and diffusivity. This type of 1D percolating morphology may have important implications for creating ordered nano-porous materials for emerging applications[58].

## Methods

### Fluoride salt synthesis

Two recipes of salt were used in the corrosion experiments for different purposes: (1) FLiNaK with 5 wt% EuF₃; (2) FLiNaK without EuF₃.
(1)  FLiNaK with 5 wt% EuF₃
      This kind of salt was used for all molten salt corrosion experiments except for the one in Fig. 2a, b. The salt has a composition of 27.75 wt% LiF, 11.11 wt% NaF, 56.14 wt% KF, and 5 wt% EuF₃. The ratio between LiF, NaF, and KF forms a eutectic (FLiNaK), while EuF₃ was added to increase the redox potential of the salt. Here, EuF₃ served as the oxidant with a reduction product of EuF₂[59]. Both EuF₃ and EuF₂ have sufficiently high solubility within FLiNaK such that neither is expected to precipitate and deposit on the

metal sample. The individual salt powders were purchased from Alfa Aesar with certified purities of 99.99%, except for $EuF_3$ which has a certified purity of 99.98%. Before mixing, the powders were baked/melted at 900–1000 °C in glassy carbon crucibles (HTW Germany) inside of an argon atmosphere-controlled glove box with oxygen and moisture continuously measured below 1 ppm. Then the salt mixture was made by weighing the corresponding powder components and melting at 700 °C for 12 h to ensure mixing. 3.5 g of salt was used in each corrosion experiment.

(2) FLiNaK

To demonstrate that the addition of $EuF_3$ is not the reason for this 1D wormhole corrosion morphology, we performed a different corrosion experiment in the FLiNaK-only molten salt environment. The corresponding results in Fig. 2a, b show that this 1D wormhole morphology still exists.

## Fluoride salt purity

The FLiNaK is not actively purified with $HF/H_2$ or pure metal, but it is produced in a procedure where most of the water can be removed by melting individual components (at 900–1000 °C) first. All the melting is conducted in an argon-atmosphere glove box where oxygen and moisture content is continuously maintained and measured below 1 ppm. The individual salt components are all 99.99% pure, but could contain some metallic impurities such that the FLiNaK is more corrosive in the short run.

Oxygen can exist in the atmosphere where the melting of the salt or the conduction of the experiments occurs. If the experiment is performed inside of the glove box, then oxygen is less than 1 ppm. If the experiment is performed inside of the high vacuum chamber, the oxygen content can be even less than that in the glove box. Note that there can exist some oxygen in the salt in the form of oxides.

## Metal sample preparation

The model 80Ni-20Cr wt% alloy was produced by Sophisticated Alloys Inc. with a certified purity level of 99.95%. The Incoloy 800H was purchased from Metalmen Sales, Inc. The Ni-20Cr and Incoloy 800H were rolled into 30- and 27-µm-thick foils by the H. Cross Company, respectively. The 25-µm-thick 316 L stainless steel foil was purchased from Metalmen Sales, Inc. Disks of 22 mm in diameter were sectioned from each foil and used as the corrosion samples (Supplementary Fig. 1b). They were compressed by two sealing flanges (Supplementary Fig. 1a) in the corrosion facility, forming a liquid-tight seal to keep the molten salt inside. The sample area subjected to corrosion had a diameter of 14 mm. Aside from the sample, all the materials in contact with the molten salt were made of commercially pure nickel (Nickel 200/201) so as not to add any metal impurities to the salt.

The CrMnFeCoNi high-entropy alloy (HEA) was argon arc-melted 3 times and cold rolled to 80%. After that, the sample was cut into small pieces and homogenized at 1300 °C for 2 h.

The Ni-4Al sample was prepared by alloying ultra-high purity Ni with minor amounts of Al. The resulting alloy was hot forged at 900 °C, solution anneal (SA) heat treated at 950 °C for 1 h, and subsequently water quenched. The final alloy chemistry can be found in previous work[60]. The surface was polished using colloidal silica (nominally 50 nm particle size) prior to exposure.

## Molten salt corrosion experiments

Unlike most corrosion experiments where samples are submerged in the corrosive environment and corrode from all faces (i.e., a 3D exposure to corrosion), we designed a unique corrosion cell similar to a permeation test device, as illustrated in Supplementary Fig. 1a. This device uses a thin metal foil as the testing sample, where only one side is exposed to the molten salt. Such a device has enabled us to characterize the penetration of salt more systematically. In particular, the foil was thin enough so that we could use an ion-mill to prepare a *dry*

*and mechanically unperturbed* cross-section, critical to avoid the side-effects of liquid-based polishing methods that could wash out the salt in the voids or introduce undesired reactions during sample preparation. Corrosion of Ni-20Cr and 316 L stainless steel in $EuF_3$-containing FLiNaK were carried out in a high vacuum chamber ($<1.3 \times 10^{-3}$ Pa). Corrosion of Incoloy 800H in $EuF_3$-containing FLiNaK occurred in the same glove box used for salt synthesis. Both environments ensure ultra-low measured concentrations of moisture and oxygen, the presence of which would have seriously impacted the corrosion process via impurity-driven corrosion[61]. The moisture and oxygen levels in the high vacuum are lower than the glove box atmosphere. Corrosion of Ni-20Cr in FLiNaK-only salt was performed in a static submerged fashion, where corrosion attack was from both sides. The Ni-20Cr foil and the salt were loaded into a glassy carbon crucible, and corroded inside the same glove box. All molten salt corrosion experiments were operated at 650 °C for 4 h. The temperature we used is typical for expected normal operation of molten salt reactors and concentrated solar power plants, and mechanistically lies in the regime where intergranular diffusion dominates over bulk diffusion.

## Oxidation experiments

The CrMnFeCoNi high-entropy alloy was aged at 1000 °C for 168 h (a week) followed by 24 h at 900 °C, 24 h at 800 °C, 24 h at 700 °C and 24 h at 600 °C sequentially in a vacuum furnace at a pressure of ~$10^{-4}$ Pa.

Exposure of the Ni-4Al binary alloy was performed in recirculating pressurized water autoclave systems capable of operation at 330 °C at an H partial pressure at the Ni/NiO stability line (11 $cm^3$/kg). Highly polished metal coupons (colloidal silica final polish) were exposed for 100 h. Flow rate through the four-liter, stainless steel autoclave was maintained at 120 mL/min giving ~2 autoclave water exchanges per hour. Water chemistry was controlled through a combination of a boric acid and lithium hydroxide saturated mixed bed resin demineralizer, active stock solution injection, and a continuously circulating four-liter water column used to bubble gasses through the autoclave feedwater.

## Preparation of cross-sections for analysis

After each corrosion experiment, the sample foil was cut free from the chamber along the sealing edge. The foil was then sectioned into two halves (Supplementary Fig. 1c), prior to cross-section polishing performed using an argon-beam cross-section polisher (JEOL SM-Z04004T) with a voltage of 6 kV (Supplementary Fig. 1d). This liquid-free method is critical to preserve the salt contained within the voids, avoiding post-test alteration of visible results. A notch about 1 mm in width (Supplementary Fig. 1e) was polished under a beam current between 150 µA and 180 µA for around 1 h. This flat and well-polished cross-section was used for subsequent electron microscopy characterization.

## Sample preparation and SEM characterization in a FIB-SEM dual-beam system

TEM sample preparation and SEM characterization were performed in a Thermo Fisher (previously FEI) Helios G4 dual-beam system with Ga as the ion beam. The SEM images were collected with an electron beam energy of 5 keV. The relationship between the imaging direction with respect to the sample geometry is illustrated in Supplementary Fig. 1f.

The TEM samples were lifted out from the cross-section surface (See Supplementary Fig. 1e). A typical FIB lift-out process is shown in Supplementary Fig. 7. First, we used the electron beam (1.6 nA) to deposit some small patterns of Pt near the region of interest as fiducial markers (see Supplementary Fig. 7c). These markers have two roles: (1) they enhance the milling accuracy during the thinning process so that the region of interest will not be over-milled; (2) they were used to test whether there were any alignment issues during the deposition

process. After that, we deposited a Pt cap on top of the region of interest using the electron beam (5 keV, 1.6 nA). Then another Pt cap layer on top of the previous Pt cap was deposited using a Ga ion beam (30 keV, 90 pA). These Pt caps were used to protect the surface structure of the sample during the preparation process. The thin film was lifted out after making a bulk trench and U-cuts, then attached to the V-notch of the copper FIB half-grid (Omniprobe®), before being welded by Pt using ion beam (30 keV, 7 pA). The sample was then thinned by a Ga ion beam with subsequently lower energies of 30, 16, 5, 2, and 1 keV step by step on both sides, and the beam currents were changed accordingly so as only to remove the milling features from the previous step. Note that only part of the lamella was thinned such that the two edges were still thick enough to provide mechanical support, to avoid film bending during the thinning process (Supplementary Fig. 7h). The EDX mapping in Fig. 2c shows that surface was well-protected by the Pt layer after the final thinning.

The FIB-SEM 3D tomography data were collected using the FEI Auto Slice & View 4.1 software and a slice thickness of 50 nm. After data collection, the image sequence was first processed by a subroutine written in ImageJ (now renamed FIJI)[62] to correct the drift of the sample during data collection, and then re-scaled to correct the aspect ratio considering the incident angle of the electron beam. A machine learning program utilized the Weka Package[63] to classify the features in the images. Finally, Dragonfly software was used for the 3D reconstruction and data visualization.

## EBSD grain orientation mapping
To verify that the rugged lines in Fig. 3b are indeed GBs, we performed electron backscatter diffraction (EBSD) scans on the region in Fig. 3b. The results are shown in Fig. 3c. A FEI Strata DB235 SEM (FEI Company, Hillsboro, OR, USA) equipped with an Orientation Imaging Microscopy (OIM) system (Ametek EDAX, Mahwah, NJ, USA) was used to collect the data with a fine step size of 0.1 μm at 20 keV.

## TEM characterization
The FEI TitanX and ThemIS were used for STEM-EDX characterization, operating at 300 keV. Both are equipped with Bruker SuperX energy-dispersive X-ray spectroscopy (EDS) detectors that enable highly efficient EDS signal collection without the need to tilt the sample to a specific angle. Each EDX map took around half an hour to acquire.

The TEAM-1 microscope at the National Center for Electron Microscopy at the Lawrence Berkeley National Laboratory was used for 4D-STEM data collection. This microscope is double-aberration corrected and operates at 300 keV. For 4D-STEM experiments, a nano-sized electron beam (~1 nm in diameter) was rastered across the sample and a nano-beam electron diffraction (NBED) pattern was collected at each real-space position of the electron beam. The sample was tilted so that the region of interest was on one of the low-index zone axes of FCC Ni-20Cr. A Gatan K3 direct electron detector with a continuum energy filter was installed for the collection of 4D-STEM data, enabling high signal-to-noise ratio and high speed (>1000 frames/s) data collection. A special bullseye-shape condenser-2 (C2) aperture was used to shape the electron beam to enhance the accuracy of lattice spacing measurements[64]. The energy filter and the bullseye aperture were key to the accurate lattice spacing measurements. A demonstration of their effects is shown in Supplementary Fig. 8. The data were collected on the same region in a selected Ni-20Cr sample after corrosion, with the upper 1/3 of the map in the DIGM zone.

The 4D-STEM experiments for Fig. 4 were performed at spot 5 with a 10 μm C2 bullseye aperture. The convergence angle was 2 mrad and the camera length was 320 mm. In our experiments, the monochromator lens setting was adjusted once the bullseye aperture was inserted, and the sample region of interest (ROI) was found. We tried to maximize the probe current until the center beam gets saturated in the detector so that the higher-order Bragg peaks can be more bright and

clearer. That said, we were operating closer to the 50–100 pA range. Note that for the 10 μm bullseye at 2 mrad convergence angle, the size of the probe on the specimen is rather large (~1–2 nm). Even though the probe current is similar to HRSTEM, the dose rate (e/A$^2$/s) is ~500 times smaller due to the large size of the probe. A 15 eV slit for the energy filter was used. The scan step size was 5 nm. The total number of nano-beam electron diffraction (NBED) patterns collected for regions 1 and 2 in Fig. 4 are 39,516 and 177,020, respectively. A scan performed over vacuum was also collected and was used to create a template for Bragg disk detection. The py4DSTEM[65] package was used to analyze the 4D-STEM dataset. The bullseye-shaped beam template collected over vacuum is shown in Supplementary Fig. 8d, and a typical NBED pattern on the Ni-Cr sample is shown in Supplementary Fig. 8e. The locations of Bragg disks were detected by matching the vacuum template to the diffraction disks (Supplementary Fig. 8f) and then used to analyze the lattice spacing. The average lattice parameter was calculated in a region far away from the DIGM zone and used as a strain-free reference region for normalization of the measured lattice parameters.

Since the EDX data and 4D-STEM data were taken by different TEMs at the same sample orientation, a Python script was used to align them by a series of scaling, rotation, image shift, and cropping operations. The alignment relies on the presence of unique features on the sample as fiducial markers.

An aberration, C$_S$-corrected ARM 200CF operated at an accelerating voltage of 200 kV was employed to collect tilt series data on the Ni-4Al sample. Data were collected utilizing an annular dark field detector using a convergence angle of 27 mR and a collection angle of 72-294 mR. The tilt series was calculated to tilt the FIB foil against the long axis of the GB at steps of 5° for 13 steps. Calculations were performed using nanocartography[66]. The 3D reconstruction was performed by a MATLAB-based simple back projection algorithm with the following constraints applied: (1) The GB is a flat plane, which seems to be a quite accurate assumption for our dataset; (2) All oxide nano-branches have the same thickness in the direction perpendicular to the GB. Tomviz[67] was used for data visualization.

## Modeling of the phase map for the dealloying process
The decrease in lattice spacing within a DIGM zone can be interpreted as a type of phase transformation induced by the changes in Cr and vacancy fractions while the crystallographic symmetry is maintained. In this vein, the whole dealloying process can be pictured in a phase map that can be calculated by DFT simulations.

For the density functional theory (DFT) simulations, the initial structure of a Ni-Cr alloy with 19.4% Cr atomic fraction was generated in a cell with 72 atoms. Cr atoms were removed from the cell to introduce vacancies. The equilibrium volumes of the cell with different Cr and vacancy fractions were calculated through conjugate-gradient minimization of the energy with respect to ionic positions, cell volume, and cell shape. Energy and force calculations during relaxation were performed using the Projector Augmented Wave (PAW)[68] method with spin-polarization considered, as implemented in the Vienna Ab-Initio Simulation Package (VASP)[69–71]. A plane wave cut-off energy of 400 eV was employed, and the Brillouin zone integrations were performed using Monkhorst–Pack meshes[72] with a 4 × 4 × 4 grid. Atomic positions were relaxed with a convergence criterion for the forces of 0.02 eV/Å. Use was made of the Perdew–Burke–Ernzerhof generalized-gradient approximation (GGA) for the exchange-correlation function[73]. Five different cells with increasing vacancy fractions were created by removing the Cr atoms one by one. After relaxation in VASP, the vacancy formation volume was calculated through linear fitting of the equilibrium volumes of cells. The lattice constant change due to vacancies without any replacement of Ni can thus be calculated based on the vacancy formation volume, as shown by the blue line in Supplementary Fig. 9. For the case where Cr fraction change involved replacement of Cr with Ni atoms, without any vacancy formation, the

lattice parameter was fit using experimental data as shown in Supplementary Fig. 10. From these experimental data we can thus obtain the eigen-volume associated with a Cr atom being replaced by a Ni atom.

With the DFT-calculated vacancy formation volume and the fitted eigen-volume for replacement of Cr by Ni derived from Supplementary Fig. 10, we can compute the relationship between lattice parameter and Cr fraction at any fixed replacement fraction as shown in Supplementary Fig. 9 through linear interpolation. Based on Supplementary Fig. 9, by incorporating the vacancy fraction contour, we can further obtain a detailed phase map as shown in Fig. 4b, which describes the intrinsic relationship between the relaxed lattice parameter $a_0$, the Cr fraction $C_{Cr}$, and the vacancy fraction $C_v$ in the relaxed state.

### Monte Carlo simulations of grain boundary migration

A 2D Monte Carlo simulation code was written in MATLAB to visualize the effects of grain boundary migration on the salt penetration speed. This code was based on the following assumptions:

(1) The initial GB is perpendicular to the bulk-metal/salt interface. The salt on the left is etching the GB and trying to penetrate to the other side of the metal. (Supplementary Fig. 2).

(2) When DIGM is enabled, a limited range (called the "dynamical DIGM region") ahead of the corrosion front will undergo grain boundary migration. A random force field will be applied to the dynamical DIGM region to deform the GBs.

(3) The GBs, once filled with salt, will turn from blue to red. The migration of the red GBs is prohibited. Note that the GBs may still migrate when they are filled with salt, but we do not consider this effect in our model.

## Data availability

The data that supports the findings of this study are available from the corresponding author upon request.

## Code availability

py4DSTEM is an open-source package available on GitHub: https://github.com/py4dstem/py4DSTEM.

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

## Acknowledgements

Primary support for this work came from FUTURE (Fundamental Under-standing of Transport Under Reactor Extremes), an Energy Frontier Research Center funded by the U.S. Department of Energy (DOE), Office

of Science, Basic Energy Sciences (BES). Y.Y., S.Y., Q.Y., and R.O.R. were supported by the Director, Office of Science, Office of BES, Materials Sciences and Engineering Division, of the U.S. DOE under Contract No. DE-AC02-05-CH11231 within the Mechanical Behavior of Materials (KC 13) program at the Lawrence Berkeley National Laboratory (LBNL). Y.Y. was also supported by the National Science Foundation (NSF) early career award DMR-2145455 after he moved to the Pennsylvania State University (PSU). The authors acknowledge support by the Molecular Foundry at LBNL, which is supported by the U.S. DOE under Contract No. DE-AC02-05-CH11231. Characterization of the Ni-Al sample was supported by the U.S. DOE, Office of Science, BES, Materials Sciences and Engineering Division, Mechanical Behavior and Radiation Effects program under FWP 56909 at Pacific Northwest National Laboratory (PNNL). PNNL is operated for the U.S. DOE by Battelle Memorial Institute under Contract No. DE-AC05-76RLO1830. J.L. acknowledges support by DOE Office of Nuclear Energy, Nuclear Energy University Program (NEUP) under Award Number DE-NE0008751. W.Z., M.J., and M.P.S. gratefully acknowledge funding from the US DOE NEUP under Grant No. 327075-875J. S.E.Z. was supported by STROBE, a NSF Science and Technology Center. S.Y.W. was supported by the NSF Graduate Research Fellowship (No. DGE 1752814). J.C. acknowledges additional support from the Presidential Early Career Award for Scientists and Engineers (PECASE) through the U.S. DOE. The authors thank Dr. Hamish Brown from the National Center for Electron Microscopy (NCEM), Molecular Foundry, LBNL and Prof. Clive Randall from the Materials Research Institute at the PSU for helpful discussions.

## Author contributions

Y.Y. and W.Z. conceived the project. A.M.M., J.L., M.P.S., M.A., and R.O.R. provided critical guidance on the project. Y.Y. performed all FIB-SEM-based experiments, TEM sample preparation and electron microscopy characterization, and carried the Monte Carlo simulation. J.C. provided guidance on the energy-filtered 4D-STEM experiments. W.Z. designed the molten salt corrosion cell, carried the corrosion experiment, and prepared the cross-sections by $Ar^+$ ion milling. Y.Y., A.M.M., and S.Y. conceived the vacancy mapping method. S.Y. performed the DFT simulations. Y.Y. and S.Y.W. performed the 4D-STEM data analysis with the help of S.E.Z. Y.Y. performed the analysis of the FIB-SEM 3D tomography result with the help of Y.Z. and M.C.S. Q.Y. performed the EBSD analysis. M.L., M.J., and J.R.S. provided important feedback that facilitated a systematic data analysis. Y.Y. performed the HEA oxidation experiment. M.J.O. performed the Ni4Al TEM characterization. Y.Y., S.Y.W., W.Z., A.M.M., J.L., M.P.S., and D.K.S. wrote the manuscript. Y.Y., S.Y.W., and W.Z. plotted the figures. All authors contributed to the discussion of the results.

## Competing interests

The authors declare no competing interests.
