## [Peer Review File · Nature Communications]

One Dimensional Wormhole Corrosion in MetalsEditorial Note: This manuscript has been previously reviewed at another journal that is not operating a transparent peer review scheme. This document only contains reviewer comments and rebuttal letters for versions considered at *Nature Communications*.

REVIEWERS' COMMENTS

Reviewer #1 (Remarks to the Author):

This will be quite a short review, as the article has been transferred to Nature Communications after its initial review, in which I participated.

The article hasn't changed structurally, and still has a bit of what I would call slightly bombastic language, but the main concerns have been addressed. I still have a slight concern that some of the phenomena observed (and this would include DIGM) are just "there", rather than having some key causal role as insisted by the authors. I don't suppose they will want to insert something to that effect, but I'm sure a lot of readers in the corrosion world will have that reaction.

I would like to insist on one thing (and apologies to the authors if I missed it in the new version) - it seems to me that this kind of corrosion, in molten salts, is closely tied up with the exposure temperature. For a NiCr alloy, 650 deg C is just about the temperature where ordinary lattice diffusion starts to modify the dealloying mechanism. In fact, I know of a study that is in the proof stage with the Journal of the Electrochemical Society, in which precisely this kind of transition from typical dealloying to a more 1-D or slot-like type of penetration occurs somewhere in the 500-700 deg C range (the material was different, but still fcc). Such a key role of temperature, independent of other variables, is worthy of mention, even if the article in question is not citeable at this point.

Reviewer #2 (Remarks to the Author):

I am generally happy with how the authors have addressed my comments. There remains some controversy, shared by the other reviewers on the name chosen to label the observed corrosion (1D wormhole corrosion) and differentiate it from other forms already reported (e.g. internal oxidation, dealloying,...). The justifications provided by the authors in their responses are satisfactory, in my opinion. I'd suggest that more of the text used in their answers to address our questions is used in the manuscript text, since I can see readers having similar questions.

The vacancy mapping approach used by the authors has a tremendous potential and could justify the publication on its own. Therefore, I am happy to recommend this manuscript for publication.

Itemized list of response to reviewers' remarks

(Blue: Reviewer's comments; Black: Our response)

Reviewer: 1

Comment 1:

This will be quite a short review, as the article has been transferred to Nature Communications after its initial review, in which I participated.

The article hasn't changed structurally, and still has a bit of what I would call slightly bombastic language, but the main concerns have been addressed. I still have a slight concern that some of the phenomena observed (and this would include DIGM) are just "there", rather than having some key causal role as insisted by the authors. I don't suppose they will want to insert something to that effect, but I'm sure a lot of readers in the corrosion world will have that reaction.

Response.

We would like to thank Review #1 for providing helpful comments. We are glad to know that the main concerns have been addressed.

Comment 2:

I would like to insist on one thing (and apologies to the authors if I missed it in the new version) - it seems to me that this kind of corrosion, in molten salts, is closely tied up with the exposure temperature. For a NiCr alloy, 650 deg C is just about the temperature where ordinary lattice diffusion starts to modify the dealloying mechanism. In fact, I know of a study that is in the proof stage with the Journal of the Electrochemical Society, in which precisely this kind of transition from typical dealloying to a more 1-D or slot-like type of penetration occurs somewhere in the 500-700 deg C range (the material was different, but still fcc). Such a key role of temperature, independent of other variables, is worthy of mention, even if the article in question is not citeable at this point.

Response.

Thank you for suggesting a new and relevant publication. After searching, we think the paper written by Ghaznavi, Persaud and Newman¹ is the paper you mentioned. It has been published and we are happy to cite this paper in our manuscript to point out the key role of temperature on the corrosion morphology, as we fully agree that the temperature is a critical factor. In fact, in our previous manuscript, we have mentioned that “While we have first identified this mechanism as occurring in a molten salt environment, one might envision similar scenarios in other dissolution-driven systems with the right balance of dissolution rates and diffusivity.” We think temperature is one of the critical factors that tunes the balance of dissolution rates and diffusivity.

Also, we would like to comment that the “1D” corrosion that Ghaznavi et al. observed is a kind of “2D” corrosion based on the definition used in our manuscript. The comparisons between the 1D

wormhole corrosion in our paper and that in their paper are shown in Fig. R1 below. For the 1D corrosion in our paper, the void will be discontinuous (just like isolated dots) along GBs in a cross-section. In the paper by Ghaznavi et al., the voids look like long and continuous lines in the cross-sectional view, which is significantly different from our work. We assume that the 3D morphology of this slot-like voids in Ghaznavi et al.'s paper will look like platelet rather than 1D tunnels. Thus, according to our definition, the corrosion morphology in their paper is a kind of “2D” corrosion. In addition, the 1D wormholes are along GBs, while those shown in Ghaznavi et al.'s paper (indicated by white arrows by the authors) are mostly intragranular voids. We have introduced Ghaznavi et al.'s paper in the revised manuscript, as shown in Fig. R2. The above discussions are also added in the Supplementary Material.

Figure R1. Difference in the corrosion morphology between our work and previous work by Ghaznavi *et al*¹.

As a consequence of 1D corrosion, strategies must now be developed to slow, prevent, or stop wormhole-type corrosion in order to increase the longevity and safety of next-generation nuclear reactors and concentrated solar power plants. The deep infiltration of salt into metal also indicates that the corrosion front is hidden deep inside the bulk, where the local chemistry can be significantly different from that near the bulk surface. As such, modeling and experimental efforts should account for the unique local chemistry inside the wormholes in order to better understand their corrosion behavior and protect against it. Also, temperature has recently been found to play a critical role on the corrosion morphologies in molten salt⁵⁹. Understanding how temperature interacts with 1D wormhole corrosion, especially the temperature range where 1D wormhole dominates, would offer valuable insights into the mitigation of it in structural materials. Last but not least, corrosion has been recently utilized as a method to produce unique void structures, such as 1D nanotubes⁶⁰ and 3D bi-continuous structures^{34,61}, for functional applications such as catalysis or sensors. While we have first identified this mechanism as occurring in a molten salt environment, one might envision similar scenarios in other dissolution-driven systems with the right balance of dissolution rates and diffusivity. This new type of 1D percolating morphology may have important implications for creating ordered nano-porous materials for emerging applications⁶².

Figure R2. Revision of the manuscript.

Interestingly, the one channel marked by a yellow circle (added by us) in the Fig. R1 rather than the ones indicated by the white arrows (added by Ghaznavi et al) looks similar to our 1D wormhole corrosion morphology. In the previous round of responses to reviewers, we showed that we could create morphologies that are similar to the morphologies shown by Ghaznavi et al in Fig. R1, if we increase the temperature or the concentration of the susceptible element in the alloys.

Reviewer: 2

Comment 1:

I am generally happy with how the authors have addressed my comments. There remains some controversy, shared by the other reviewers on the name chosen to label the observed corrosion (1D wormhole corrosion) and differentiate it from other forms already reported (e.g. internal oxidation, dealloying,...). The justifications provided by the authors in their responses are satisfactory, in my opinion. I'd suggest that more of the text used in their answers to address our questions is used in the manuscript text, since I can see readers having similar questions.

The vacancy mapping approach used by the authors has a tremendous potential and could justify the publication on its own. Therefore, I am happy to recommend this manuscript for publication.

Response.

We appreciate the positive and encouraging comments of Review #2 again, especially for pointing out that *“The vacancy mapping approach used by the authors has a tremendous potential and could justify the publication on its own.”*

We also like the suggestion of adding more texts from our point-by-point reply to the manuscript. We have added the discussions to Supplementary materials file, pages 6-9. A short discussion has also been added in the main text, as shown in the figure below.

or etched precipitates³⁷. However, 1D corrosion is differentiated by the fact that the voids are interlinked and form a percolating network. Filiform corrosion^{38,39} can also produce similar 1D and percolated channels, but these surface effects are confined to specific metal/organic-film interfaces, and therefore coupled with a thin film delamination process. The 1D-like channel formation underneath the organic coatings during filiform corrosion is not considered a bulk corrosion (*i.e.*, depth penetration) mechanism like 1D wormhole corrosion. Also, the 1D wormhole corrosion is significantly different from other “1D” corrosion morphologies reported in previous literatures, with more discussions in the Supplementary Discussion.

Pore structure is a distinguishing feature of localized versus uniform corrosion, as it

Figure R3. Revision of the manuscript.

References

1. Ghaznavi, T., Persaud, S. Y. & Newman, R. C. The Effect of Temperature on Dealloying Mechanisms in Molten Salt Corrosion. *J. Electrochem. Soc.* **169**, 111506 (2022).